# Assessment of the Impact of Lower Urinary Tract Dysfunction on Quality of Life in Multiple Sclerosis Patients in Saudi Arabia—A Cross-Sectional Study

**DOI:** 10.3390/healthcare11192694

**Published:** 2023-10-09

**Authors:** Mansour Abdullah Alghamdi, Khaled Abdulwahab Amer, Abdulrahman Ali S. Aldosari, Reemah Farhan Al Qahtani, Haneen Saeed Shar, Lujane Mohammed Al-Tarish, Rammas Abdullah Shawkhan, Mohammad Ali Alahmadi, Mohammed Abadi Alsaleem, Laith Naser AL-Eitan

**Affiliations:** 1Department of Anatomy, College of Medicine, King Khalid University, Abha 62529, Saudi Arabia; 2Genomics and Personalized Medicine Unit, College of Medicine, King Khalid University, Abha 62529, Saudi Arabia; 3Internal Medicine Department, Aseer Central Hospital, Abha 62523, Saudi Arabia; 4Department of Radiology, Aseer Central Hospital, Abha 62523, Saudi Arabia; 5College of Medicine, King Khalid University, Abha 62529, Saudi Arabia; 6Department of Physical Education and Sport Sciences, College of Education, Taibah University, Madinah 42353, Saudi Arabia; 7Department of Family and Community Medicine, College of Medicine, King Khalid University, Abha 62529, Saudi Arabia; 8Department of Applied Biological Sciences, Jordan University of Science and Technology, Irbid 22110, Jordan; 9Department of Biotechnology and Genetic Engineering, Jordan University of Science and Technology, Irbid 22110, Jordan

**Keywords:** lower urinary tract dysfunction, quality of life, healthcare surveys, multiple sclerosis, Saudi Arabia

## Abstract

Background: Lower urinary tract dysfunction (LUTD) is caused by neurogenic factors that could lead to permanent injury in affected patients, and therefore result in substantial annual healthcare expenses. LUTD is very prevalent in multiple sclerosis (MS) patients and has a drastic impact on their quality of life (QOL). This study aimed to assess the effect of LUTD on the QOL of Saudi MS patients. Methods: A cross-sectional study was carried out in Saudi Arabia using a self-administered questionnaire that included the World Health Organization Quality of Life (WHOQOL-BREF) and LURN Symptom Index (LURN SI-29). Data were analyzed and presented as frequencies and percentages. Results: There were 428 patients who participated in this study; 270 were females and 158 were males. Most of the patients received a low score in all sections of the LURN part of the questionnaire. The highest scores (urgent need to urinate and excessive urination at night) were recorded in the urgency domain (47.20 ± 36.88) rather than the nocturia domain (44.74 ± 32.91). Meanwhile, the lowest score (complete control of bladder) was recorded in the incontinence domain (22.80 ± 26.80). For the WHOQOL-BREF score, the highest score (more social stability) was in the social domain (65.07 ± 21.16 for females, 60.41 ± 21.54 for males), and the lowest score (less psychological stability) was in the psychological domain (46.36 ± 9.84 for females, 46.20 ± 10.03 for males). However, there was no significant association between the four domains of the WHOQOL-BREF and the gender of the MS patients. Conclusions: LUTD is significantly associated with a lowered quality of life. Therefore, patients are recommended to consult with and be evaluated by appropriately experienced healthcare providers and clinicians. This ensures that the patients receive the best advice, accurate and effective treatment, and long-term analysis that can lead to an improvement in their quality of life.

## 1. Introduction

Multiple sclerosis (MS) is a frequent central nervous system (CNS) illness that often appears in early adulthood and varies depending on the origin of inflammation [1,2,3,4]. MS is divided into three categories: The most frequent subtype of MS is relapsing–remitting MS, accounting for roughly 85% of patients, and is distinguished by acute neurological symptoms with intervals of recovery [5,6]. The primary progressive form (present in 15% of patients) is characterized by gradual and progressive neurological symptoms with few flares [5]. The most prevalent autonomic disruption in persons with MS is neurogenic lower urinary tract dysfunction (LUTD), which continues to be a clinical challenge to treat, resulting in significant yearly healthcare expenses and causing significant handicaps in afflicted persons that affect many elements of daily living [2,7,8,9].

Urinary difficulties are present in 35% of individuals with multiple sclerosis (MS) at the time of diagnosis, and more than 75% have urinary issues at some point during the disease, developing voiding dysfunction and LUTD [1,9,10]. However, LUTD symptoms may be recorded in one out of every ten MS patients at the time of the initial MS presentation, while the incidence of LUTD symptoms and dysfunction grows over time, approaching 100% in 10 years because of the progressive nature of MS [9]. Urinary abnormalities can also be the first manifestation of the illness, as is seen in 2–10% of patients [11]. LUTD is best divided into three categories: urine storage (commonly known as “irritative”), voiding (sometimes known as “obstructive”), and post-micturition symptoms [12,13]. Urine storage and voiding are two distinct but linked phases of LUTD [12]. The muscle and connective tissue in the bladder and urethra develop an intrinsic tone [12]. At rest, the urethral tone maintains wall apposition and aids in incontinence [14]. The bladder walls relax and become sensitive during filling (the vesical lumen increases without causing an increase in intravesical pressure) [12]. When a specific level of fullness is achieved (which varies depending on circumstances and individuals), the growing afferent activity begins to intrude on consciousness, resulting in awareness that the bladder is filling up [12]. Other symptoms include increased daytime and nighttime (nocturia) urination frequency, urinary urgency, and incontinence, whereas voiding symptoms include an interrupted and weak stream, urinary hesitancy, straining to urinate, and the sensation of incomplete bladder emptying after voiding [6,8,13]. Even though urine incontinence has a significant impact on patients’ lives, the ailment is frequently underreported [8]. This is mainly because many patients avoid seeking medical attention since they feel ashamed and shy to share their concerns with healthcare professionals [8]. Likewise, healthcare practitioners regularly neglect to mention urine symptoms, and patients risk facing multiple difficulties (like skin infections, urinary tract infections, falls, and fractures) [8].

LUTD is often related to poor-quality sleep, low energy, poor emotional health, and a lower overall quality of life; however, it has also been associated with having a negative influence on an individual’s social, occupational, psychological, and sexual life [8,11,13,14,15,16,17]. Because all aspects of life may change and vary, a conceptual model based on the biomedical and social sciences paradigm is required to represent this variegated experience. This model should encompass the psychological, biological, and social dimensions of health and life quality [1,15,18]. Studies have shown that UTIs and community-acquired UTIs are related to a lower quality of life (QOL) and health-related QOL [19,20,21]. A study on Saudi women with community-acquired UTIs showed that different aspects of their life were impacted due to their infections, and therefore lowered their health-related QOL [21]. The study showed that UTIs negatively affected their bed rest, work and schooling, and mobility, and increased their pain [21]. Collectively, this leads to a lowered QOL [21]. Thus, this study aimed to assess the impacts of LUTD on QOL in MS patients in the Saudi Arabian population. This should provide a basis for further studies to assess the quality of healthcare institutions and medical consultations available to patients in Saudi Arabia, thus aiding in the development of the healthcare system in Saudi Arabia.

## 2. Subjects and Methods

### 2.1. Study Population and Ethics Statement

A cross-sectional study was conducted using a validated self-administered questionnaire consisting of demographic and clinical characteristics, the World Health Organization Quality of Life scale (WHOQOL-BREF), and the LURN Symptom Index (LURN SI-29). In total, 428 patients of both genders in Saudi Arabia who were diagnosed with MS and older than 18 years were invited to participate in the study. Data were collected from a survey that was disseminated through social media in collaboration with MS societies from June 2021 to January 2022. Ethical approval was obtained from the local IRB committee before data collection. The purpose and methodology of the study were explained clearly to the participants before they enrolled and provided informed consent. Provisions for protecting privacy were relevant at all stages of the research, including in subject identification, recruitment, participation, and analysis.

Sampling and less-than-true responses may be potential sources of prejudice. To reduce sampling biases, the questionnaires were distributed through MS social media accounts and MS associations to improve visibility and attract more respondents. To address the problem of bias in the responses, we created an anonymous self-administration questionnaire and avoided leading questions and answers.

### 2.2. Measurement Scale

#### WHOQOL-BREF and LURN Scale

The WHOQOL-BREF is a 26-item scale that was created in eighteen countries to be applied as a general assessment of QOL [22]. It is a viable tool to assess QOL across cultures and illnesses [22]. It is grouped into four domains: physical health, psychological health, social ties, and environmental health [22]. Therefore, the questions (Q) in this study were divided as follows: physical/physiological health (9 questions), psychological health (6 questions), social ties (3 questions), and environmental health (8 questions) [22].

The LURN Symptom Index-29 (LURN SI-29) was created as a patient-reported outcome (PRO) measure to evaluate urinary symptoms in both genders (covers 27 questions) [23]. The LURN SI-29 is intended for use by those who have urinary symptoms [18]. The LURN SI-29 and WHOQOL-BREF were employed in our investigation, where 428 people completed the study questionnaire. Of those, 270 (63.10%) were females and 158 (36.90%) were males. Nearly 72.4% of the participants were between the ages of 20 and 40. The LURN SI-29 covers 27 questions for both men and women, as well as two sex-specific questions (1 for women and 1 for men) [18]. The LURN SI-29 subscale and overall scores are standardized on a scale of 0 to 100 [23]. The LURN SI-29 measures incontinence (Q1–Q6), bladder pain (Q6–Q10), voiding difficulties (Q11–Q15), urgency (Q16–Q18), nocturia (Q19–Q20), frequency (Q21–Q26), and post-micturition symptoms over the last 7 days, as well as asking two sex-specific questions (1 for women and 1 for men) [18,23]. The tests can be given on paper or online [23]. Scoring the LURN SI-29 is accurate when all questions are answered; hence, 100% completion should be encouraged.

To maintain the effectiveness and efficiency of the questionnaire and verify its psychometric validity, the questionnaire document was translated into Arabic by a certified independent translator, who conducted forward and backward translations. In addition, the validity and reliability of the translated questionnaire were examined by a medical sociology professor using face validity. Before the distribution of the questionnaire, pilot testing was conducted to evaluate its reliability and validity.

### 2.3. Statistical Analysis

The information gathered was processed and provided in the form of frequencies and percentages. The chi-squared test, Pearson’s correlation, and Fisher’s exact test were used to test the quantitative data, and the differences between means were determined using IBM SPSS Statistics software for Windows, version 26.0 (IBM Corp., Armonk, NY, USA). In all analyses, two-sided *p*-values less than 0.05 were judged statistically significant. The data were reviewed for missing data and outliers before analysis. Means and standard deviations are used to represent continuous variables. Outliers were calculated using the interquartile range (IQR) method. In brief, the data were sorted from low to high, and then the first quartile (Q1), median, and third quartile (Q3) were identified. This was followed by calculating the IQR (Q3 − Q1), and then calculating the upper fence (Q3 + (1.5 × IQR) and lower fence (Q1 − (1.5 × IQR)). Finally, any values that fell outside the fences were considered outliers.

## 3. Results

The study questionnaire was completed by 428 people who were asked to participate regardless of the occurrence of LUTD between June 2021 and January 2022, and their answers were assessed using the WHOQOL-BREF and LURN SI-29 scales. There were 270 females (63.10 percent) and 158 men (36.90 percent) who participated. Patients were classified into five age groups based on their age; 72.4 percent of the patients were between the ages of 20 and 40. There were three illiterate participants, and therefore they were unable to fill out their questionnaires by themselves. Thus, the questionnaire was read to them out loud by the data collection contributor, and they were asked to answer by choosing one of the answers verbally. Other participant demographic variables and scale scores are presented in Table 1.

As shown in Table 2 and Appendix A, the majority of the patients mostly had a low score in all sections of the LURN portion of the questionnaire; i.e., the symptoms of LUTD have little or few effects on the MS patients. They are able to completely control their bladder during their different life activities (incontinence part); have either no or little pain, or are comfortable during the urination process (bladder pain); never or a few times have difficulties during urination (voiding difficulties part); never or a few times have an urgent need to urinate; and never or a few times have post-micturition symptoms. Furthermore, the total score of the LURN sections was significantly associated with gender (*p*-value < 0.05), whereas sections A and E were exceptions. 

Regarding the bladder pain section, Q.9 and Q.10 had a *p*-value of 0.000326 and 0.033, respectively, which indicates that the severity degree of the pain is increased in men. For the voiding difficulty section, 49.3% of females compared to 26.6% of males in the past 7 days never had a delay before they started to urinate and 3.7% of females and 8.2% of males had a delay every time before starting to urinate (Q.12). Once urinating starts, the frequency that the urine flow stops and starts again (Q.13), having a urine flow that is slow or weak (Q.14), and having a dribble or trickle at the end of the urine flow (Q.15) is increased in men compared to women. Nearly 60% of males compared to 40% of females feel a sudden need to urinate about half of the time, most of the time, and every time (Q.16). Female patients more frequently experience post-micturition problems compared to males (Q. 25 and 26).

The highest score was obtained for section D (urgency domain) (47.20 ± 36.88) rather than section E (nocturia domain) (44.74 ± 32.91), which was the same for both genders, and the lowest score was for section A (incontinence domain) (22.80 ± 26.80), which was also the same for both genders, as shown in Table 3. Male patients reported high scores (*p*-value < 0.05) in the bladder pain, voiding difficulty, urgency, and post-micturition domains compared with female patients. Boxplots of the subscale scores and the total scores of the LURN SI-29 of both sexes are shown in Figure 1, indicating that women reported lower scores for all domains, and men reported the highest scores for urgency and nocturia compared to women. Overall, it appears that the average total LUTS and the variability of scores indicate that the burden is higher for men compared to women (LURN SI-29 total median score = 25.29 for women, 30.91 for men, variance 316.97 vs. 241.6, *p* < 0.001).

For the WHOQOL-BREF scores, as are presented in Table 4 and Appendix A, and Figure 2, the highest score was in the social domain for both genders (65.07 ± 21.16 for females, 60.41 ± 21.54 for males), and the lowest score was in the psychological domain (46.36 ± 9.84 for females, 46.20 ± 10.03 for males). Although there was no significant association between the four domains of the WHOQOL-BREF part of the questionnaire and the gender of the Saudi Arabian MS patients (*p*-value > 0.05), there were six questions out of the twenty six displaying significant differences between both genders, as demonstrated in Table 4: Q1 (*p*-value = 0.043), Q4 (*p*-value = 0.007), Q13 (*p*-value = 0.038), Q15 (*p*-value = 0.008), Q21 (*p*-value = 0.002), and Q22 (*p*-value = 0.011).

The median scores of the WHOQOL-BREF were generally the same for both genders for the subscales and the total scores, and no significant difference was observed between male and female scores (Figure 2); i.e., the quality of life of UTI MS patients was not affected by gender, and so they had the same scores on this questionnaire.

## 4. Discussion

Urinary abnormalities are an initial indicator of LUTD in 2% to 10% of MS patients, and they are present from the start of the disease [10]. A recent study in the United States showed that uncomplicated UTIs (the most common UTIs) severely cause mobility impairments [24]. Consequently, UTIs affect the health-related QOL and productivity of female patients [24]. LUTD is found quite frequently in MS patients (75%) and it is sometimes cited as the disease’s worst symptom [25], as it includes voiding and urine retention symptoms (25%), storage symptoms (50%), and/or a combination of symptoms (25%) [25]. Failure to empty the bladder is caused by detrusor-distal sphincter dyssynergia or areflexia. Furthermore, urgent incontinence is common for both sexes and is associated with voiding trouble and having decreased sensation [1]. These symptoms not only create a major psychosocial burden, but also necessitate medical intervention and hospitalization, and present a significant obstacle for the care team [1,10,25]; in addition, they have a considerable detrimental influence on MS patients’ quality of life [9,11,13,25,26].

This study aimed to measure the effect of LUTD on the QOL of Saudi MS patients. Several studies have been conducted to investigate the negative impact of LUTD on daily living activities and health-related QOL in people with MS [6]. Urine symptoms in MS have a detrimental influence on QOL as they restrict involvement in daily activities, negatively influence emotional status, social roles, and employment, and can lead to an increased risk of falls in older MS patients [7]. The QOL of MS is influenced not only by urinary symptoms and other health-related complications, but other factors such as social life, physical activity, and environment can play a critical role. Therefore, measuring the effect of LUTD on the QOL of MS patients in the context of different populations and cultures can give different insights into the issue. The results of our study confirmed that improving these factors can help to improve the QOL of MS patients.

Patients should inquire about the following symptoms with the clinician: frequency, urgency, urge or stress incontinence, bladder discomfort, hematuria, dysuria, or retention [23]. The duration, intensity, and level of annoyance should all be established. A review of the patient’s surgical and medical history may identify disease processes that are directly connected to their concerns [27]. Accurate drug determination is particularly critical because many medicines might have a direct influence on lower urinary tract function [22,23]. A comprehensive obstetric and menstrual history should be collected for female patients. Standardized validated surveys are great instruments for identifying symptoms, their level of bother, and their influence on QOL. More than 30 questionnaires in English have been made available to assess neurogenic bladder symptoms and their influence on a patient’s QOL [28].

This study revealed, through asking incontinence-related questions, that a significant number of both male and female patients mostly have control over their bladder during different life activities (Table 2 and Appendix A). When patients where asked about bladder pain or discomfort and voiding difficulty, a majority of them had no or only a few instances of pain/discomfort and voiding difficulty during the urination process (Table 2 and Appendix A). From the urgency section (Q16–Q18), it was found that about 33.30% to 39.60% of both male and female patients never felt a sudden need to urinate or difficulty in waiting more than a few minutes, whereas 32.90% (Q16) and 34.80% (Q18) of male patients felt a sudden need to urinate about half of the time and found it somewhat difficult to wait more than a few minutes before urinating (Table 2 and Appendix A). According to the nocturia section, about 31.50% and 32.90% of female and male patients, respectively, wake up one time and urinate during a typical night. Moreover, 35.90% of female patients wake up at least once a few nights in a week and 31% of male patients wake up at least once about half of the nights in a week to urinate. About 40% of female and 50% of male patients urinate 4–7 times in a typical day during waking hours. Moreover, 58% of female and 57% of male patients need 3–4 h between urination. Additionally, about 43% and 42% of female and male patients, respectively, felt a few times that their bladder was not completely empty after urination. Also, a majority of patients never dribble urine after zipping their pants or pulling up their underwear. About 48% of female patients and 44% of male patients never experienced spraying or splitting of the urine stream. When patients were asked about urinary symptoms bothering them, about 39% of females and 49% of males said they were somewhat bothered.

We found that MS male patients reported higher scores in all domains in the LURN SI-29 (Table 3), indicating that they were more affected by LUTD compared to females (mean total scores: 25.29 and 30.91 for females and males, respectively) with a *p*-value < 0.05. Males reported higher bladder pain, voiding difficulty, urgency, and post micturition compared to females (*p*-value < 0.05). A recent study has shown that bladder management improved the QOL of MS female patients with UTIs [29]. This was determined through consultations that involved bladder training programs and Kegel exercises over a 3-month period [29].

In this study, the highest score was in the social domain for both genders (65.07 ± 21.16 for females, 60.41 ± 21.54 for males) (Figure 2), and the lowest score was in the psychological domain (46.36 ± 9.84 for females, 46.20 ± 10.03 for males). Although there was no significant association between the four domains of the WHOQOL-BREF and the gender of the Saudi Arabian MS patients (*p*-value > 0.05), six questions showed significant differences between both genders including QOL rate (Q1), medical treatment needed to function in their daily life (Q4), the availability of information that they need in day-to-day life (Q13), the ability to get around (Q15), sex life satisfaction (Q21), and the support they get from friends (Q22). In the physical/physiological domain, when patients were asked about their sleep satisfaction, ability to perform daily living activities, and capacity for work, their responses ranged from neither satisfied nor dissatisfied to satisfied. About 46% (female) and 44% (male) of patients reported that physical pain can moderately prevent them from doing what they need to do. Moreover, 38% (female) and 32% (male) of patients reported moderate energy for everyday life. In the psychological domain, 42% (female) and 42% (male) of patients enjoy life to a moderate amount. Interestingly, about 35% to 39% of patients felt that their life is meaningful to an extreme amount. Moreover, about half of the patients were able to moderately concentrate. The highest percentage of both female and male patients’ reported responses that ranged from neither satisfied/dissatisfied to satisfied occurred when they were asked questions about their social life (such as personal relationship, sex life, and friends’ support). In the environmental domain, our results showed that about 31% of patients feel safe in their daily life to a moderate amount, and 37% claimed that their physical environment is moderately healthy. Additionally, the highest percentage of patients were satisfied with the conditions of their living place, as well as with their access to health services and transportation.

Urinary difficulties in MS patients vary due to the multifocal pattern of MS lesions and the disease’s history of numerous exacerbations [10,30]. Using urodynamic tests to identify the kind of bladder-sphincter dysfunction is crucial for guiding therapy selection and avoiding upper urinary tract abnormalities [10,30]. One essential objective of treating urinary troubles in MS patients is to reduce complications and improve their health-related QOL (HRQOL) [10].

Self-report bias and participants’ capacity to adequately record their sickness condition, drugs, symptoms, and therapy are potential study limitations that come with this survey technique. Another constraint is selection bias. This convenience sample consisted of patients who self-selected to join in this study and who were required to have an Internet connection; therefore, they may have had characteristics that are not shared by the overall MS population. Patients in the later stages of the disease, for example, may have been less interested in completing a questionnaire due to weariness, mobility issues, or the ability to concentrate on a computer screen for long periods. In addition, there is poor communication among healthcare professionals and fewer referrals of people with MS to specialist services due to a lack of understanding of these specialized services. Another further limitation is that the results of the current study do not provide generalized results regarding MS patients worldwide. However, it is a step in the right direction for further understanding the state and quality of life of MS patients. This could eventually help in providing sustainable solutions for MS patients to increase their quality of life and improve their physiological, psychological, and social well-being. A longitudinal study to further assess the QOL of Saudi MS patients with LUTD is suggested. This could further justify and clarify the impact of these diseases on the QOL of Saudi patients and provide a basis for a comparison of the development of healthcare systems and social acceptance between Saudi Arabia and other countries.

## 5. Conclusions

LUTD is characterized by several medical symptoms, including storage, voiding, and post-micturition phases. LUTD is also associated with considerable morbidity in persons with MS, leading to mobility limitations and a worse health-related QOL. In this study, two scales were included in the questionnaire, which were used to measure QOL in MS patients with LUTD. In both, LUTD is substantially associated with a low quality of life. As a result, patients should be assessed and treated by appropriately competent healthcare consultants who can choose one of the numerous successful treatment choices available, and therefore enhance their patients’ QOL.

## Figures and Tables

**Figure 1 healthcare-11-02694-f001:**
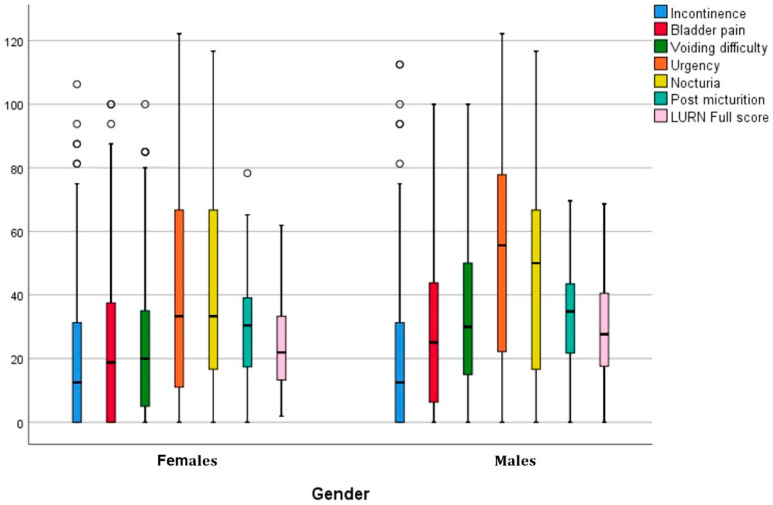
LURN SI-29 scale and subscale total scores by sex. (The line in the box and the lower and upper edges of the box indicate the median and the 25th (Q1) and 75th (Q3) percentiles, respectively. The whiskers represent the minimum (extends to Q1 − 1.5 * interquartile range) and maximum (extends to Q3 + 1.5 * interquartile range). Circles indicate outliers.) LURN SI-29, Lower Urinary Tract Dysfunction Research Network Symptom Index-29.

**Figure 2 healthcare-11-02694-f002:**
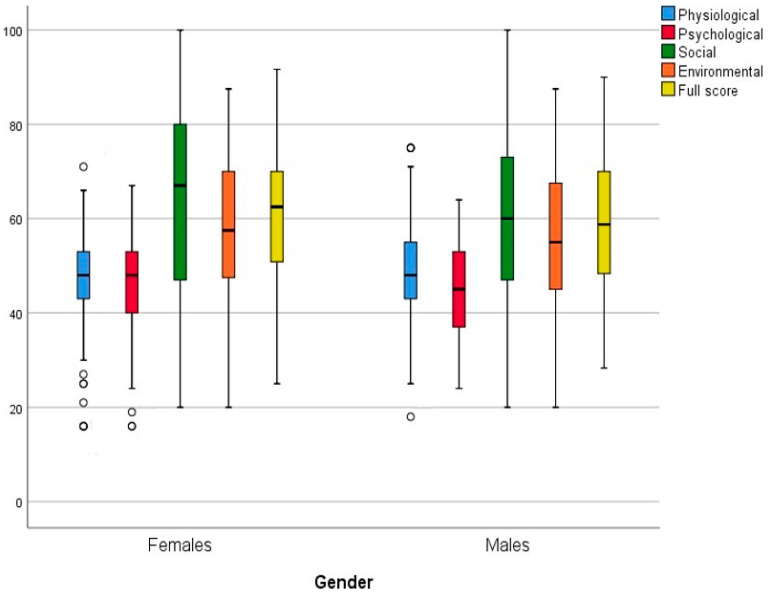
The WHOQOL-BREF scale and subscale total scores by sex (270 females and 158 males). (The line in the box and the lower and upper edges of the box indicate the median and the 25th (Q1) and 75th (Q3) percentiles, respectively. The whiskers represent the minimum (extends to Q1 − 1.5 * interquartile range) and maximum (extends to Q3 + 1.5 * interquartile range). Circles indicate outliers.).

**Table 1 healthcare-11-02694-t001:** Sociodemographic characteristics of MS patients who responded to the WHOQOL-BREF and LURN scale questionnaire.

Parameters		Gender				Participants (428)	
		Females (270, 63.1%)		Males (158, 36.9%)			
		Count	%	Count	%	Count	%
Age	Less than 20	9	2.1%	1	0.2%	10	2.3%
	20–30	103	24.1%	51	11.9%	154	36.0%
	31–40	94	22.0%	62	14.5%	156	36.4%
	41–50	53	12.4%	37	8.6%	90	21.0%
	Older than 50	11	2.6%	7	1.6%	18	4.2%
Educational level	High school	46	10.7%	41	9.6%	87	20.3%
	Illiterate	2	0.5%	1	0.2%	3	0.7%
	Secondary school	14	3.3%	5	1.2%	19	4.4%
	University	208	48.6%	111	25.9%	319	74.5%
Marital status	Divorced	26	6.1%	7	1.6%	33	7.7%
	Married	120	28.0%	84	19.6%	204	47.7%
	Single	123	28.7%	67	15.7%	190	44.4%
	Widow	1	0.2%	0	0.0%	1	0.2%
Occupation	Homemaker	99	23.1%	0	0.0%	99	23.1%
	Retired	12	2.8%	24	5.6%	36	8.4%
	Student	34	7.9%	18	4.2%	52	12.1%
	Teacher	25	5.8%	14	3.3%	39	9.1%
	Other	100	23.1%	102	23.8%	201	47.0%
MS type	Primary Progressive	27	6.3%	9	2.1%	36	8.4%
	Relapsing–Remitting	78	18.2%	44	10.3%	122	28.5%
	Secondary Progressive	7	1.6%	17	4.0%	24	5.6%
	Unknown	158	36.9%	88	20.6%	246	57.5%
Family history of MS	No	230	53.7%	136	31.8%	366	85.5%
	Yes	40	9.3%	22	5.1%	62	14.5%

**Table 2 healthcare-11-02694-t002:** The significant LURN questionnaire scores of the participants.

Questionnaire	Frequency	Patients		Gender				*p*-Value *
				Females		Males		
		Count	%	Count	%	Count	%	
Q9. In the past 7 days, how often did you have pain or discomfort while urinating?	never	204	47.70%	143	53.00%	61	38.60%	0.000326
	a few times	132	30.80%	87	32.20%	45	28.50%	
	about half the time	38	8.90%	15	5.60%	23	14.60%	
	most of the time	34	7.90%	17	6.30%	17	10.80%	
	every time	20	4.70%	8	3.00%	12	7.60%	
Q10. In the past 7 days, how often did you have pain or discomfort right after you had finished urinating?	never	250	58.40%	171	63.30%	79	50.00%	0.033
	a few times	97	22.70%	59	21.90%	38	24.10%	
	about half the time	32	7.50%	14	5.20%	18	11.40%	
	most of the time	34	7.90%	18	6.70%	16	10.10%	
	every time	15	3.50%	8	3.00%	7	4.40%	
Q12. In the past 7 days, how often did you have a delay before you started to urinate?	never	175	40.90%	133	49.30%	42	26.60%	0.00004
	a few times	141	32.90%	84	31.10%	57	36.10%	
	about half the time	40	9.30%	20	7.40%	20	12.70%	
	most of the time	49	11.40%	23	8.50%	26	16.50%	
	every time	23	5.40%	10	3.70%	13	8.20%	
Q13. In the past 7 days, once you started urinating, how often did your urine flow stop and start again?	never	183	42.80%	128	47.40%	55	34.80%	0.000301
	a few times	134	31.30%	84	31.10%	50	31.60%	
	about half the time	45	10.50%	32	11.90%	13	8.20%	
	most of the time	43	10.00%	16	5.90%	27	17.10%	
	every time	23	5.40%	10	3.70%	13	8.20%	
Q14. In the past 7 days, how often was your urine flow slow or weak?	never	165	38.60%	115	42.60%	50	31.60%	0.003
	a few times	147	34.30%	99	36.70%	48	30.40%	
	about half the time	50	11.70%	26	9.60%	24	15.20%	
	most of the time	39	9.10%	19	7.00%	20	12.70%	
	every time	27	6.30%	11	4.10%	16	10.10%	
Q.15 In the past 7 days, how often did you have a trickle or dribble at the end of your urine flow?	never	143	33.40%	106	39.30%	37	23.40%	0.000096
	a few times	131	30.60%	85	31.50%	46	29.10%	
	about half the time	48	11.20%	30	11.10%	18	11.40%	
	most of the time	59	13.80%	23	8.50%	36	22.80%	
	every time	47	11.00%	26	9.60%	21	13.30%	
Q16. In the past 7 days, how often did you feel a sudden need to urinate?	never	124	29.00%	90	33.30%	34	21.50%	0.003
	a few times	93	21.70%	66	24.40%	27	17.10%	
	about half the time	105	24.50%	53	19.60%	52	32.90%	
	most of the time	72	16.80%	40	14.80%	32	20.30%	
	every time	34	7.90%	21	7.80%	13	8.20%	
Q25. In the past 7 days, how often did you feel that your bladder was not completely empty after urination?	never	120	28.00%	64	23.70%	57	36.10%	0.008
	a few times	182	42.50%	116	43.00%	66	41.80%	
	about half the time	39	9.30%	23	8.50%	16	10.10%	
	most of the time	58	13.70%	43	15.90%	15	9.50%	
	every time	28	6.50%	24	8.90%	4	2.50%	
Q26. In the past 7 days, how often did you dribble urine just after zipping your pants or pulling up your underwear?	never	238	55.60%	148	54.80%	90	57.00%	2.04 × 10^−7^
	a few times	123	28.70%	74	27.40%	49	31.00%	
	about half the time	28	6.50%	15	5.60%	13	8.20%	
	most of the time	26	6.10%	24	8.90%	2	1.30%	
	every time	13	3.00%	9	3.30%	4	2.50%	
Q28. In the past 7 days, how bothered were you by urinary symptoms?	Not at all bothered	123	28.70%	76	28.10%	47	29.70%	0.041
	Somewhat bothered	183	42.80%	106	39.30%	77	48.70%	
	Very bothered	59	13.80%	39	14.40%	20	12.70%	
	Extremely bothered	63	14.70%	49	18.10%	14	8.90%	

* *p*-value < 0.05 is considered statistically significant.

**Table 3 healthcare-11-02694-t003:** The total section scores of the LURN SI-29.

Section		Patients		Gender				*p*-Value *
				Females (270)		Males (158)		
		Mean	SD	Mean	SD	Mean	SD	
Section A	Incontinence	22.80	26.80	21.75	26.23	24.58	27.76	0.2920
Section B	Bladder pain	26.05	25.64	23.58	24.59	30.28	26.89	0.0090
Section C	Voiding difficulty	28.40	24.59	24.13	22.49	35.70	26.32	0.0000
Section D	Urgency	47.20	36.88	43.54	36.68	53.45	36.51	0.0070
Section E	Nocturia	44.74	32.91	42.96	31.72	47.79	34.75	0.1430
Section F	Post micturition	31.16	15.95	29.39	14.87	34.17	17.26	0.0003
Full score		27.37	16.62	25.29	15.54	30.91	17.80	0.0010

* *p*-value < 0.05 is considered statistically significant.

**Table 4 healthcare-11-02694-t004:** The significant WHOQOL-BREF questionnaire scores of the participants.

Questionnaire	Frequency	Patients		Gender				*p*-Value *
				Females		Males		
		Count	%	Count	%	Count	%	
Q1. How would you rate your quality of life?	Very poor	15	3.5%	9	3.3%	6	3.8%	
	Poor	38	8.9%	23	8.5%	15	9.5%	
	Neither poor nor good	186	43.5%	104	38.5%	82	51.9%	0.043
	Good	132	30.8%	96	35.6%	36	22.8%	
	Very good	57	13.3%	38	14.1%	19	12.0%	
Q4. How much do you need any medical treatment to functionin your daily life?	Not at all	67	15.7%	36	13.3%	31	19.6%	
	A little	69	16.1%	33	12.2%	36	22.8%	
	A moderate amount	136	31.8%	91	33.7%	45	28.5%	0.007
	Very much	24	5.6%	17	6.3%	7	4.4%	
	An extreme amount	132	30.8%	93	34.4%	39	24.7%	
Q13. How available to you is the information that you need in your day-to-day life?	Not at all	33	7.7%	21	7.8%	12	7.6%	
	A little	83	19.4%	41	15.2%	42	26.6%	
	Moderately	112	26.2%	78	28.9%	34	21.5%	0.038
	Mostly	107	25.0%	66	24.4%	41	25.9%	
	Completely	93	21.7%	64	23.7%	290	18.4%	
Q15. How well are you able to get around?	Very poor	32	7.5%	19	7.0%	13	8.2%	
	Poor	50	11.7%	24	8.9%	26	16.5%	
	Neither poor nor good	90	21.0%	48	17.8%	42	26.6%	0.008
	Good	116	27.1%	81	30.0%	35	22.2%	
	Very good	140	32.7%	98	36.3%	42	26.6%	
Q21. How satisfied are you with your sex life?	Very dissatisfied	70	16.4%	38	14.1%	32	20.3%	
	Dissatisfied	70	16.4%	34	12.6%	36	22.8%	
	Neither satisfied nor dissatisfied	126	29.4%	80	29.6%	46	29.1%	0.002
	Satisfied	108	25.2%	82	30.4%	26	16.5%	
	Very satisfied	54	12.6%	36	13.3%	18	11.4%	
Q22. How satisfied are you with the support you get from your friends?	Very dissatisfied	72	16.8%	47	17.4%	25	15.8%	
	Dissatisfied	58	13.6%	35	13.0%	23	14.6%	
	Neither satisfied nor dissatisfied	107	25.0%	54	20.0%	53	33.5%	0.011
	Satisfied	113	26.4%	83	30.7%	30	19.0%	
	Very satisfied	78	18.2%	51	18.9%	27	17.1%	

* *p*-value < 0.05 is considered statistically significant.

## Data Availability

Datasets associated with the paper were analyzed and are presented as results or embedded tables in the paper.

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
