# Peer review of "Assessment of the Impact of Lower Urinary Tract Dysfunction on Quality of Life in Multiple Sclerosis Patients in Saudi Arabia—A Cross-Sectional Study"

_healthcare, 2023, doi:10.3390/healthcare11192694_

Round 1
Reviewer 1 Report
Introduction: Authors can add a few lines about understanding the impact of UTI on QOL and how can it contribute to patient care and well-being
Methods: Cross-sectional design has a limitation in establishing the casual association of disease in interest. Longitudinal studies would provide a better understanding of UTI and QOL association in MS.
Discussion: A better explanation of the results in the discussion section is warranted.
Author Response
Comments by Reviewer 1
Introduction: Authors can add a few lines about understanding the impact UTI on QOL and how can it contribute to patient care and well-being.
Response: The issue has been addressed in the introduction section (last paragraph).highlighted in yellow
Methods: Cross-sectional design has a limitation in establishing the casual association of disease in interest. Longitudinal studies would provide a better understanding of UTI and QOL association in MS.
Response: The authors would like to state that a longitudinal study could be done in the future to further strengthen the results of the current study. Also, a suggestion has been added in the last paragraph of discussion section.highlighted in yellow
Discussion: A better explanation of the results in the discussion section is warranted.
Response: detailed explanation of the results was added in the discussion section (paragraph 4-6).highlighted in yellow
Reviewer 2 Report
This work looked at survey response from 428 patients with multiple sclerosis (MS) in Saudi Arabia. Survey instruments included validated surveys for measuring quality of life and lower urinary tract symptoms, in addition to sociodemographic characteristics. The authors present data on the various domains of the surveys and compare responses of males versus females.
Overall, the presence of lower urinary tract issues was associated with poorer quality of life. This is not surprising, but having data on the various domains impacted by lower urinary tract dysfunction may help patients and healthcare providers have more specific conversations which may lead to improved treatment.
Strengths:
- Sample size of more than 400 patients
- Use of validated surveys
- Wide sampling method
Weaknesses
- It is difficult to understand how the results fit into the context of what is already known about urinary problems in MS. The discussion spends too much time reviewing results that have already been presented and not enough time telling the reader what is new or different about this study.
- The tables of the survey results (Tables 2 and 4) are not readable. The authors should consider finding other ways to display the data. Perhaps with bar charts. Perhaps only display data from questions that are were statistically significant and have the remainder in supplemental material.
- Please discuss how outliers were identified in the methods section. Some of the data points in Figures 1 and 2 do not look like they should be considered outliers.
- You can remove the patient numbers from the outliers in Figures 1 and 2 as they are not helpful or informative.
- The abstract would be improved by saying whether a low score (line 31) is good or bad. Same for the high score on line 33. If readers are not familiar with these survey tools they will not be able to understand the significance of a high or low score.
- Details information on the survey tools (lines 228-230 and 242-243) should be moved to the methods section, rather than be in the discussion section.
- Line 135, please change “girls” to “females”, as girls refer to a young age.
- In Table 1, three participants are listed as “Illiterate”, meaning they cannot read or write. Do you know how they were able to complete the survey in that case?
The paper could benefit from English editing for clarity, sentence construction, and use of certain language.
Author Response
Comments by Reviewer 2
This work looked at survey response from 428 patients with multiple sclerosis (MS) in Saudi Arabia. Survey instruments included validated surveys for measuring quality of life and lower urinary tract symptoms, in addition to sociodemographic characteristics. The authors present data on the various domains of the surveys and compare responses of males versus females.
Overall, the presence of lower urinary tract issues was associated with poorer quality of life. This is not surprising, but having data on the various domains impacted by lower urinary tract dysfunction may help patients and healthcare providers have more specific conversations which may lead to improved treatment.
Strengths:
- Sample size of more than 400 patients
- Use of validated surveys
- Wide sampling method
Weaknesses
- It is difficult to understand how the results fit into the context of what is already known about urinary problems in MS. The discussion spends too much time reviewing results that have already been presented and not enough time telling the reader what is new or different about this study.
Response: discussed your point in the discussion section (second paragraph).highlighted in green
- The tables of the survey results (Tables 2 and 4) are not readable. The authors should consider finding other ways to display the data. Perhaps with bar charts. Perhaps only display data from questions that are were statistically significant and have the remainder in supplemental material.
Response: The tables have been amended according to the reviewer’s suggestion and a supplementary file (Table S1 and Table S2) has been added.
- Please discuss how outliers were identified in the methods section. Some of the data points in Figures 1 and 2 do not look like they should be considered outliers.
Response: added details about calculating the outliers in section 2.3 statistical analysis. highlighted in green
- You can remove the patient numbers from the outliers in Figures 1 and 2 as they are not helpful or informative.
Response: patient numbers were removed from outliers in figures 1 and 2.
- The abstract would be improved by saying whether a low score (line 31) is good or bad. Same for the high score on line 33. If readers are not familiar with these survey tools they will not be able to understand the significance of a high or low score.
Response: Thank you for your valuable comment, a definition of low and high score was added to the abstract and put between two brackets. highlighted in green
- Details information on the survey tools (lines 228-230 and 242-243) should be moved to the methods section, rather than be in the discussion section.
Response: Details information on the survey tools has been moved to section 2.2.1 WHOQOL-BREF and LURN Scale. highlighted in green
- Line 135, please change “girls” to “females”, as girls refer to a young age.
Response: The issue has been amended.
- In Table 1, three participants are listed as “Illiterate”, meaning they cannot read or write. Do you know how they were able to complete the survey in that case?
Response: They were asked verbally by the data collection contributor and answered by choosing one of the available options. This has been clarified in the reviewed manuscript (result section, first paragraph). highlighted in green
Comments on the Quality of English Language
The paper could benefit from English editing for clarity, sentence construction, and use of certain language.
Response: English language has been proofread using MDPI English service (certificate is available upon request).
Reviewer 3 Report
This is an interesting manuscript tackling the impact of lower urinary tract disorder on patients from Saudi Arabia with multiple sclerosis. There is a need for certain modification before their paper can be accepted.
First, a meticulous and exhaustive revision of the English language is needed. This imperative step is driven by the notable prevalence of erroneous sentence constructions, which not only impair the clarity and coherence of the text, but also hinder its overall effectiveness in conveying the intended message. Additionally, a significant issue arises where sentences initiate with numerical values, a practice that deviates from standard linguistic conventions and further contributes to the text's lack of fluidity and correctness.
In the Methods part of the manuscript, SPSS should be properly cited: IBM SPSS Statistics for Windows, version 26.0 (IBM Corp., Armonk, N.Y., USA). It should be stated whether p-value was two-tailed; this clarification is vital as it offers transparency regarding the analytical approach and the interpretation of significance levels.
Frequencies and percentages are not considered qualitative data like the authors state; they are types of quantitative data. Qualitative data involves descriptions and characteristics that cannot be easily quantified, such as opinions, emotions, or textual responses. Quantitative data, on the other hand, involves measurable quantities and numerical values that can be subjected to mathematical analysis. Frequencies and percentages fall under the category of quantitative data as they represent counts or proportions of occurrences within a dataset (the authors probably meant to say categorical data).
Some other limitations should be mentioned. For example, recall bias has not been mentioned, and findings might not be easily generalizable beyond the specific context of the study or the sample population. Also, the Discussion section should include more recent studies from the field, as there has been important developments. (i.e., only one study from 2021 and two studies from 2022 are cited, and no studies from 2023).
English should be improved by a native English speaker in order to increase the value of the manuscript.
Author Response
Comments by Reviewer 3
This is an interesting manuscript tackling the impact of lower urinary tract disorder on patients from Saudi Arabia with multiple sclerosis. There is a need for certain modification before their paper can be accepted.
First, a meticulous and exhaustive revision of the English language is needed. This imperative step is driven by the notable prevalence of erroneous sentence constructions, which not only impair the clarity and coherence of the text, but also hinder its overall effectiveness in conveying the intended message.
Response: English language has been proofread using MDPI English service (certificate is available upon request).
Additionally, a significant issue arises where sentences initiate with numerical values, a practice that deviates from standard linguistic conventions and further contributes to the text's lack of fluidity and correctness.
Response: The issue has been addressed.
In the Methods part of the manuscript, SPSS should be properly cited: IBM SPSS Statistics for Windows, version 26.0 (IBM Corp., Armonk, N.Y., USA).
Response: The issue has been addressed in section 2.3 statistical analysis. highlighted in Blue
It should be stated whether p-value was two-tailed; this clarification is vital as it offers transparency regarding the analytical approach and the interpretation of significance levels.
Response: two-sided p-value was used. More information has been added to the method section 2.3. statistical analysis. highlighted in Blue
Frequencies and percentages are not considered qualitative data like the authors state; they are types of quantitative data. Qualitative data involves descriptions and characteristics that cannot be easily quantified, such as opinions, emotions, or textual responses. Quantitative data, on the other hand, involves measurable quantities and numerical values that can be subjected to mathematical analysis. Frequencies and percentages fall under the category of quantitative data as they represent counts or proportions of occurrences within a dataset (the authors probably meant to say categorical data).
Response: The issue has been addressed in section 2.3 statistical analysis. highlighted in Blue
Some other limitations should be mentioned. For example, recall bias has not been mentioned, and findings might not be easily generalizable beyond the specific context of the study or the sample population.
Response: The issue has been addressed and added details in discussion section (last paragraph). highlighted in Blue
Also, the Discussion section should include more recent studies from the field, as there has been important developments. (i.e., only one study from 2021 and two studies from 2022 are cited, and no studies from 2023).
Response: More recent studies have been included in the discussion (Reference 24 and 29).highlighted in Blue
Comments on the Quality of English Language
English should be improved by a native English speaker in order to increase the value of the manuscript.
Response: Response: English language has been proofread using MDPI English service (certificate is available upon request).
Round 2
Reviewer 2 Report
Thank you for making the suggested revisions to improve the quality of the manuscript.